# Machine-learning-accelerated design of high-performance platinum intermetallic nanoparticle fuel cell catalysts

Peng Yin[1,3], Xiangfu Niu[2,3], Shuo-Bin Li[1], Kai Chen[2], Xi Zhang[1], Ming Zuo[1], Liang Zhang [2] ✉ & Hai-Wei Liang [1] ✉

Carbon supported PtCo intermetallic alloys are known to be one of the most promising candidates as low-platinum oxygen reduction reaction electrocatalysts for proton-exchange-membrane fuel cells. Nevertheless, the intrinsic trade-off between particle size and ordering degree of PtCo makes it challenging to simultaneously achieve a high specific activity and a large active surface area. Here, by machine-learning-accelerated screenings from the immense configuration space, we are able to statistically quantify the impact of chemical ordering on thermodynamic stability. We find that introducing of Cu/Ni into PtCo can provide additional stabilization energy by inducing Co-Cu/Ni disorder, thus facilitating the ordering process and achieveing an improved tradeoff between specific activity and active surface area. Guided by the theoretical prediction, the small sized and highly ordered ternary $Pt_2CoCu$ and $Pt_2CoNi$ catalysts are experimentally prepared, showing a large electrochemically active surface area of ~90 $m^2$ $g_{Pt}^{-1}$ and a high specific activity of ~3.5 mA $cm^{-2}$.

Proton exchange membrane fuel cells (PEMFCs) with net-zero carbon emission are promising energy conversion devices[1,2], but the heavy use of high-cost platinum-based electrocatalysts for boosting the sluggish oxygen reduction reaction (ORR) at the cathode limits their large-scale commercialization[3,4]. To this end, the US Department of Energy (DOE) has set two mass-normalized performance based on platinum group metal (PGM) as cost targets, including rated power (>8 kW $g_{PGM}^{-1}$) and activity (>0.44 A $mg_{PGM}^{-1}$)[5]. Hence, significant reduction of PGM usage by using ORR electrocatalysts with high mass activity (MA) is imperative to achieve full-commercialization of PEMFCs[6].

Recently, carbon supported structurally ordered Pt-based intermetallic compound (IMC) nanoparticles have been extensively investigated as low-Pt catalysts to boost ORR for PEMFCs[7–10]. Since the pronounced ordering-degree-dependent activity in intermetallic catalysts, the realization of high or even full ordering degree is highly desirable when preparing the alloy catalysts[11,12]. To promote the ordering degree of IMCs catalysts, high-temperature annealing is crucial to form the Pt-M alloys with ideal stoichiometric ratio and to overcome energy barrier of disorder-to-order transition within every individual nanoparticle[13], which inevitably leads to the sintering of catalysts into larger particles with decreased electrochemical surface area (ECSA)[14,15]. As demonstrated in a typical impregnation synthesis of PtCo alloy, the seesaw relation between particle size and ordering degree could be clearly observed (Fig. 1a). An acceptable MA of large-particle catalysts with a low ECSA could be achieved by the compensation of a very high SA[16–18]. However, for a certain Pt usage in the membrane electrode assembly (MEA, $mg_{Pt}/cm^2_{MEA}$), a low ECSA means a small Pt roughness factor in MEA (normalized ECSA on a cathode, $cm^2_{Pt}/cm^2_{MEA}$) that exacerbates the local oxygen transfer resistance, which eventually leads to a significantly decreased fuel cell performance, particularly at the high current density[6,19].

[1]Hefei National Research Center for Physical Sciences at the Microscale, Department of Chemistry, University of Science and Technology of China, Hefei 230026, China. [2]Center for Combustion Energy, School of Vehicle and Mobility, State Key Laboratory of Intelligent Green Vehicle and Mobility, Tsinghua University, Beijing 100084, China. [3]These authors contributed equally: Peng Yin, Xiangfu Niu. ✉e-mail: zhangbright@tsinghua.edu.cn; hwliang@ustc.edu.cn

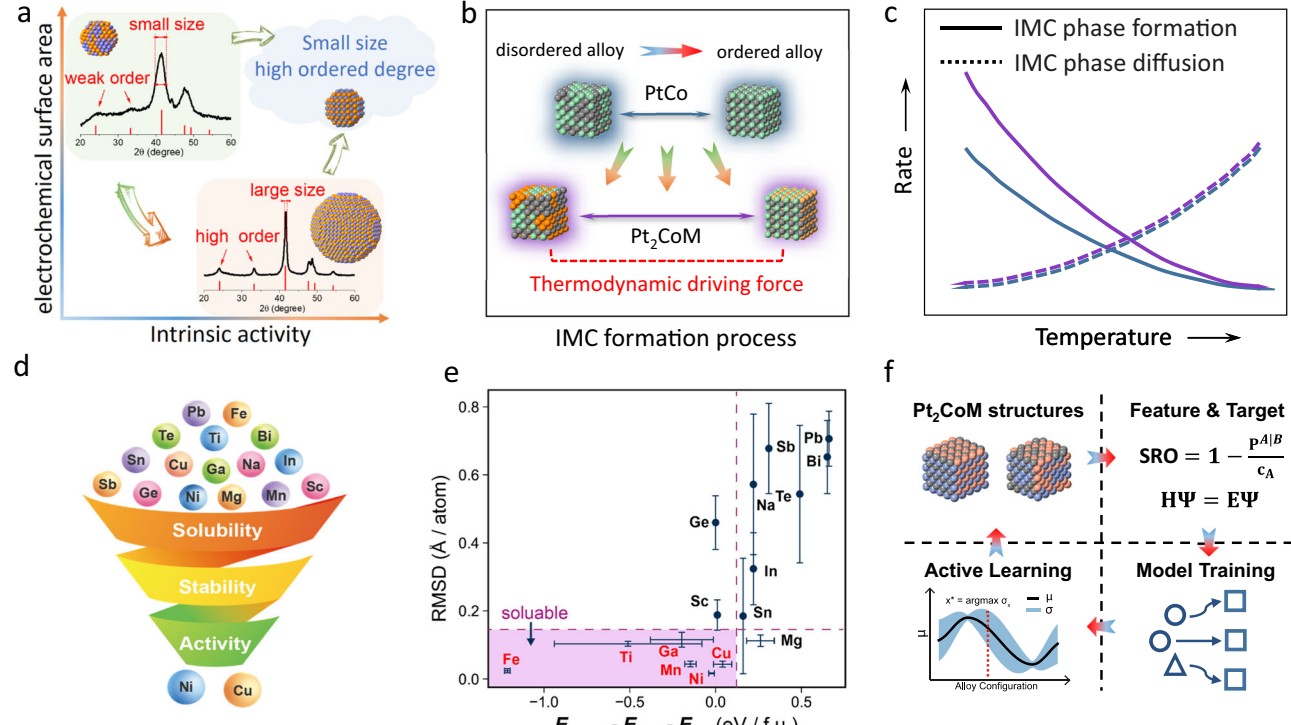

**Fig. 1 | Development of machine learning model. a** Schematic illustration showing the dilemma of ECSA and SA in a typical binary PtCo IMC catalyst synthesis. **b** Schematic illustration assuming the presence of Pt$_2$CoM combination with higher thermodynamic driving force of disordered-to-ordered transition. **c** Schematic illustration showing the change of IMC phase formation rate after enhancing thermodynamic driving force. Purple and blue colors represent Pt$_2$CoM and PtCo, respectively. **d** Screening flowchart of ternary Pt$_2$CoM combinations. **e** Structural deformation and relative energy distribution of 16 Pt$_2$CoM combinations. **f** Active learning procedures to construct machine learning prediction model for relative energy of Pt$_2$CoM.

Some elaborately designed methods have been developed for preparing small-size Pt-based intermetallic catalysts, such as KCl matrix-assisted annealing[20], low temperature chemical vapor deposition with organometallic precursors[21], sulfur-anchoring synthesis[8], small-molecule assisted synthesis[22], and thermal decomposition of bimetallic complexes[23]. These methods focus on the anti-sintering during the high-temperature annealing; the ordering degree of the resulted IMCs catalysts is often low, even though the high kinetic energy barrier of atom ordering could be overcome by the high-temperature annealing. The reason behind this phenomenon is probably the low thermodynamic driving force in the disordered-to-order transition process[13,15], which would significantly limit the nucleation rate of IMC phase.

Recently, machine learning methods have demonstrated significant potential in accelerating material discovery by efficiently navigating design spaces and predicting properties, thereby substantially reducing the cost of identifying and optimizing catalytic materials[24–27]. Here, we perform the machine-learning-accelerated computational screening, aiming at the de novo design of the element composition to increase the thermodynamic driving force for the disordered-to-ordered transition and thus promote the nucleation of IMC phase with high ordering degree (Fig. 1b, c). After the systematic screening of the ternary Pt$_2$CoM alloys (PtCo represents the most promising alloy catalyst for practical PEMFCs applications[28,29]; M is another base metal element), including alloy mutual solubility, ordering transition energy, and strain-induced activity prediction, we obtain two optimal solutions of Pt$_2$CoCu and Pt$_2$CoNi. The experimentally prepared Pt$_2$CoCu (Ni) IMC catalysts show both large ECSA of ~90 m$^2$ g$_{Pt}$$^{-1}$ and high SA of ~3.5 mA cm$^{-2}$, which lead to a high MA of ~3 A mg$_{Pt}$$^{-1}$. The highly ordered Pt$_2$CoCu catalysts also exhibit enhanced MEA performance in practical H$_2$–air fuel cells.

## Results and discussion

### Computational screening of ternary Pt$_2$CoM alloys

To reduce the experimental trials and errors, we performed theoretical screening of the Pt$_2$CoM ternary system with the third element M from an initial pool of 16 potential elements, including Na, Mg, Sc, Ti, Mn, Fe, Ni, Cu, Ga, Ge, In, Sn, Sb, Te, Pb and Bi. The alloying of the third element aims to facilitate the Pt-M ordering while maintaining, if not enhancing the surface ORR performance. Therefore, the 16 candidates underwent screening in the following three aspects: solubility of M in PtCo alloy, promotion of a more feasible disorder-to-order transition in ternary Pt$_2$CoM compared to binary PtCo, and the potential for higher ORR activity in Pt$_2$CoM (Fig. 1d). Consequently, this systematic approach successfully narrowed down the potential ternary candidates to two: Pt$_2$CoCu and Pt$_2$CoNi, which were subsequently experimentally verified.

Figure 1e shows the two matrics used to assess the solubility of M in PtCo alloy: the relative energy of Pt$_2$CoM with respect to PtCo and PtM, and the structural deformation due to the M dissolving. For M=Fe, Ti, Ga, Mn, Ni, Cu, Sc, and Ge, the segregation of ternary Pt$_2$CoM to PtCo and PtM is thermodynamically unfavorable. However, it is important to note that even when segregation is thermodynamically preferred, the formation of a ternary alloy can still be kinetically stabilized. Meanwhile, Ge and Sc exhibit significant structural deformation after the density functional theory (DFT) optimization (Fig. 1e and Supplementary Fig. S1). Therefore, six soluble element Fe, Ti, Ga, Mn, Cu, and Ni, located in the left bottom of Fig. 1e were subjected to the ordering assessment in the next stage.

The activity and durability of Pt alloy catalysts are strongly dependent on the ordering degree of Pt and Co/M[11,12]. Warren-Cowley short range order (SRO) was used to quantify the chemical ordering of the alloy system. We defined the ordering energy $E_{ordering}$ as the energy

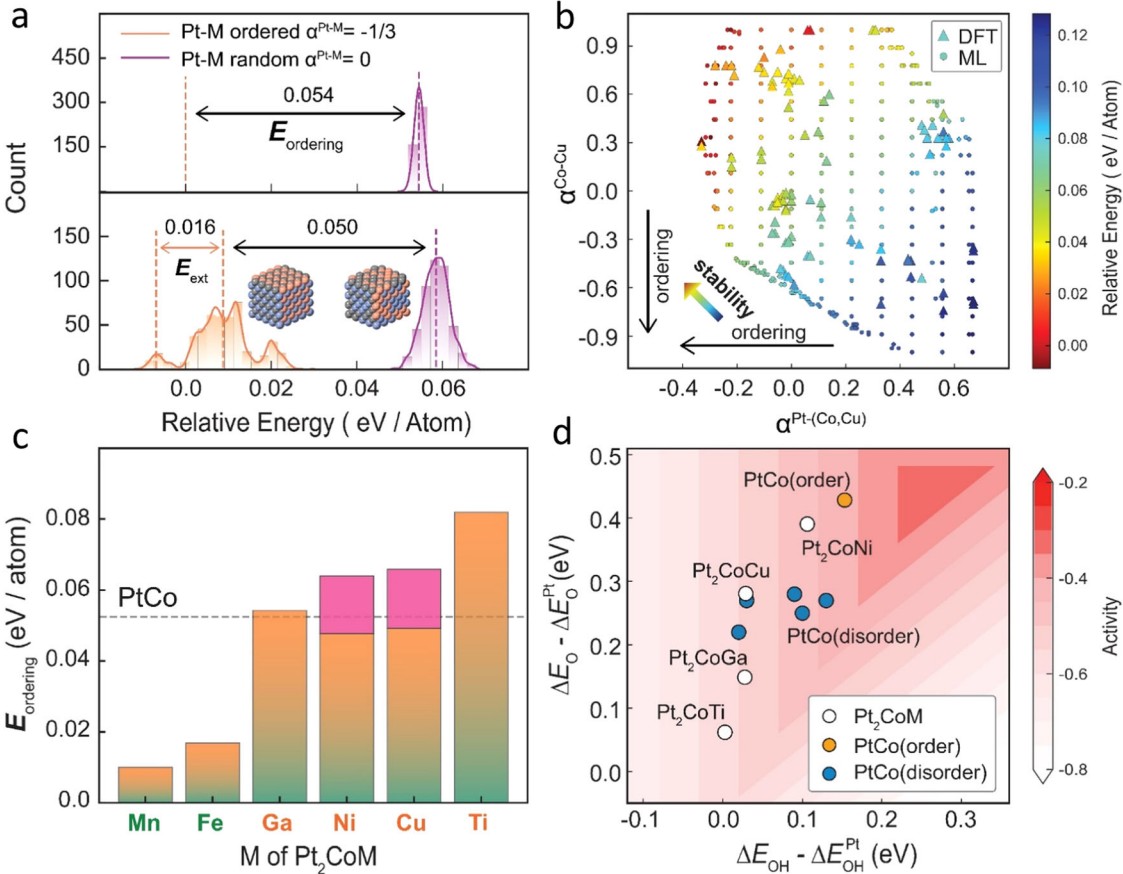

**Fig. 2 | Result feedback of machine learning model. a** ML predicted relative energies distribution of random and ordered PtCo (upper) and Pt$_2$CoCu (bottom). **b** Relative energies of Pt$_2$CoCu configurations as a function of Pt-Co/Cu SRO $\alpha^{Pt-(Co/Cu)}$ and Co-Cu SRO $\alpha^{Co-Cu}$. A higher SRO value represents a higher disordered degree. Triangles and circles represent data points computed by DFT (1 configurations per data point) and ML (average of 20 configurations for each data point) prediction model, respectively. **c** Ordering energy $E_{ordering}$ for six soluble Pt$_2$CoM combinations. Dashed line represents $E_{ordering}$ of PtCo for reference. Red part represents the extra stabilization energy from the Co/Cu disordering. **d** ORR activity volcano plot of four ordered Pt$_2$CoM, ordered PtCo and five disordered PtCo.

difference between ordered configurations (SRO of Pt-Co/M: $\alpha^{Pt-(Co/M)} = -1/3$, where Pt atoms hold the same positions as PtCo intermetallics) and randomly mixed configurations ($\alpha^{Pt-(Co/M)} = 0$). The ordering energy $E_{ordering}$ measures the thermodynamic driving force for the disorder-to-order transition. To overcome the intractability of the diverse interatomic arrangements in Pt$_2$CoM using DFT calculation, an active learning strategy was adopted to train the prediction model of relative energy for each Pt$_2$CoM ternary system (Fig. 1f and Supplementary Table S1). We selected Gaussian Process Regression (GPR) as the machine leaning model due to its ability for uncertainty measurement[24]. Active learning was implemented, and after completing 7 rounds of iterations, over 100 DFT data points were used for each Pt$_2$CoM, resulting in a formation energy prediction accuracy with an error below 10 meV/Atom (Supplementary Figs. S2–S5, Supplementary Table S2). The trained machine learning model was then applied to predict the formation energies of 300 ordered ($\alpha^{Pt-(Co/M)} = -1/3$) and 300 random ($\alpha^{Pt-(Co/M)} = 0$) configurations, as well as 3800 structures with varying ordering degrees (Supplementary Fig. S6). We found that the ordered PtCo was 0.054 eV per atom more stable than the configurations where Pt and Co were randomly mixed (upper panel of Fig. 2a). The introduction of Cu exhibits a similar thermodynamic barrier between the mean formation energy of ordered and random configurations (0.050 eV per atom). Interestingly, we found an extra small peak bringing an extra stabilization energy $E_{ext} = 0.016$ eV per atom for the Pt-Co/Cu ordered structure. Further analysis demonstrates that the extra stabilization energy of the

Pt-M ordering origins from the Co/Cu disordering (red section in Fig. 2c), where the relative formation energy of Pt$_2$CoCu increases with $\alpha^{Pt-(Co/Cu)}$, but oppositely correlates with $\alpha^{Co-Cu}$ (Fig. 2b and Supplementary Figs. S7 and S8). Similarly, we indentified that Ga, Ni, Ti held the potential to form more ordered IMC structure, but the introduction of Mn and Fe would significantly suppress the thermodynamic driving force for the disorder-to-order transition of Pt$_2$CoM (Fig. 2c and Supplementary Table S1). In the realm of computational efficiency, the training and predicting with machine learning models required negligible time compared to DFT calculations, leading to a significant reduction in the time required to establish the correlation between chemical ordering and stability (Supplementary Fig. S9).

When catalyzing ORR, the Pt alloy catalysts should be convereted into alloy@Pt core-shell structures by in situ electrochemical dealloying or pre-leaching in acid[11]. The electronic properties of the Pt-shell are strongly influenced by the alloy core due to the well-known ligand and/or strain effects[30,31]. To evaluate the ORR activity of the Pt$_2$CoM (M = Ga, Ni, Cu and Ti), we calculated the adsorption energies of OH* and O* as the activity descriptor using Pt-shell slab models with ordered Pt$_2$CoM-core (Fig. 2d and Supplementary Tables S3 and 4)[32]. The computed results show that strain effects play a dominated role in regulating the ORR activity of Pt$_2$CoM (Supplementary Fig. S10). For comparision, we also marked the calculated activity of ordered and disordered PtCo with various ordering degree of subsurface PtCo core. Generally, ordered PtCo poesses higher activity than disordered counterpart. Among the ternary alloys, Pt$_2$CoCu and Pt$_2$CoNi show

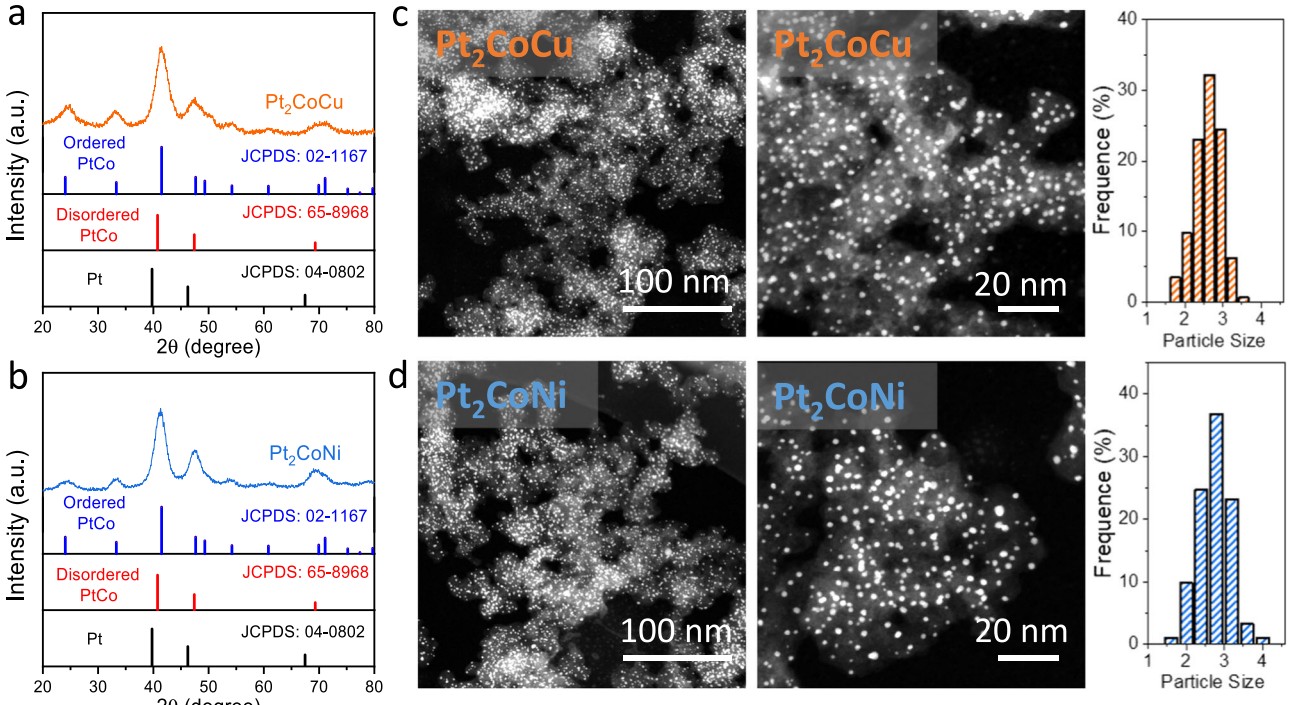

**Fig. 3 | Synthesis of Pt$_2$CoCu/Pt$_2$CoNi IMC catalysts.** XRD patterns of Pt$_2$CoCu (**a**) and Pt$_2$CoNi (**b**) catalysts. HAADF-STEM images and corresponding particle size distribution of Pt$_2$CoCu (**c**) and Pt$_2$CoNi (**d**) catalysts.

comparable ORR activity with fully ordered PtCo, and thus were finally selected for the experimental validation. There has been prior research reporting PtCoNi and PtCoCu ternary system as excellent ORR catalyst due to their near-optimum strain levels for higher ORR activity[33,34], aligning well with our computational screening results. More importantly, our study provides a innovative design perspective that the introduction of Cu/Ni leverages the thermodynamic driving force for the disordered-to-order transition, resulting in a more favorable tradeoff between specfic activity (SA) and electrochemically active surface area (ECSA).

### Synthesis and characterizations of Pt$_2$CoCu/Pt$_2$CoNi catalysts

The Pt$_2$CoCu and Pt$_2$CoNi catalysts were synthesized by the wet-impregnation of H$_2$PtCl$_6$ and corresponding non-noble metal salts on a carbon support Black Pearl 2000, followed by a high-temperature annealing at 1000 °C. We first performed the powder X-ray diffraction (XRD) characterizations to verify the L1$_0$-type intermetallic structures by comparing with the standard Powder Diffraction File cards of corresponding IMC (Fig. 3a, b). Different from a face-centered cubic structure in disordered PtCo alloy, two characteristic super-lattice peaks at around 24° and 33° were found in the Pt$_2$CoCu and Pt$_2$CoNi samples, indicating the formation of the L1$_0$ structures. After Rietveld refinement of the XRD patterns, we could obtain the fitting parameters for each diffraction peak (Supplementary Fig. S11 and Supplementary Table S5). In addition, the (111) peaks showed obvious broadening and no phase separation, suggesting a uniform IMC phase and small crystallite size. The average crystallite sizes calculated by the Debye-Scherer equation based on the full-width at half-maximum of (111) peak were 2.94 nm and 3.19 nm for Pt$_2$CoCu and Pt$_2$CoNi, respectively.

High-angle annular dark-field scanning transmission electron microscopy (HAADF-STEM) revealed that Pt$_2$CoCu/Pt$_2$CoNi IMC nanoparticles were homogeneously distributed over the whole carbon matrix and no obvious aggregates or overgrowth of nanoparticles were found in both low- and high-resolution view fields (Fig. 3c, d). Statistical analyses suggested a narrow particle size distribution of Pt$_2$CoCu/Pt$_2$CoNi catalysts in the range of 1.5–4 nm, with average size

of 2.62 and 2.73 nm, respectively, which were close to the values estimated by XRD. Further, energy dispersive spectroscopy (EDS) elemental mapping with a large visual field indicated the homogeneous distributions of Pt and other non-noble metal elements in individual nanoparticles without element segregation (Fig. 4a, b).

We then performed aberration-corrected HAADF-STEM with atomic resolution to verify the ordered structure of the Pt$_2$CoCu/Pt$_2$CoNi catalysts (Fig. 4c, d). Owing to the atomic number (Z)-contrast differences between Pt and non-noble metals in an ordered lattice, the periodic regularity of brightness could be found along the specific zone axis; the Pt atoms will appear brighter than other non-noble metals with lower Z-contrast. Through their Z-contrast and atomic radius after local amplification, we could observe an alternating bright and dark stacking of Pt and non-noble metal columns. Our theoretical predictions suggested that Co with Cu or Ni tends to exhibit random site occupation. However, distinguishing the Co/Cu or Co/Ni position in the alloy structure is challenging due to their similar atomic radius[33]. Fast Fourier transform (FFT) patterns further confirmed the face-centered tetragonal ordered structures of the Pt$_2$CoCu/Pt$_2$CoNi catalysts (Fig. 4c, d).

### Electrochemical Performance

We first evaluated the ORR activity of the catalysts by the rotating disk electrode (RDE) technique. For comparison, we also prepared binary PtCo catalysts by the identical wet-impregnation method. Two types of PtCo catalysts, including the high-ordered/large-sized one (PtCo*) and the low-ordered/small-sized one (PtCo#), were obtained at high-temperature and low-temperature annealing condition, respectively (Fig. 1a). As expected, the high-ordered/large-sized PtCo catalyst exhibited a higher intrinsic activity with SA of ~3.4 mA cm$^{-2}$ but a much lower ECSA of 15 m$^2$g$_{pt}$$^{-1}$ than that of the low-ordered/small-sized catalyst (SA: 1.67 mA cm$^{-2}$; ECSA: 87 m$^2$ g$_{pt}$$^{-1}$) (Fig. 5a and Supplementary Table S6). The trade-off relation between SA and ECSA makes the binary PtCo catalyst showing a limited mass activity (MA) of lower than 2.0 A mg$_{pt}$$^{-1}$ (Fig. 5b). In contrast, the high-ordered/small-sized ternary Pt$_2$CoCu/Pt$_2$CoNi catalysts broke such trade-off

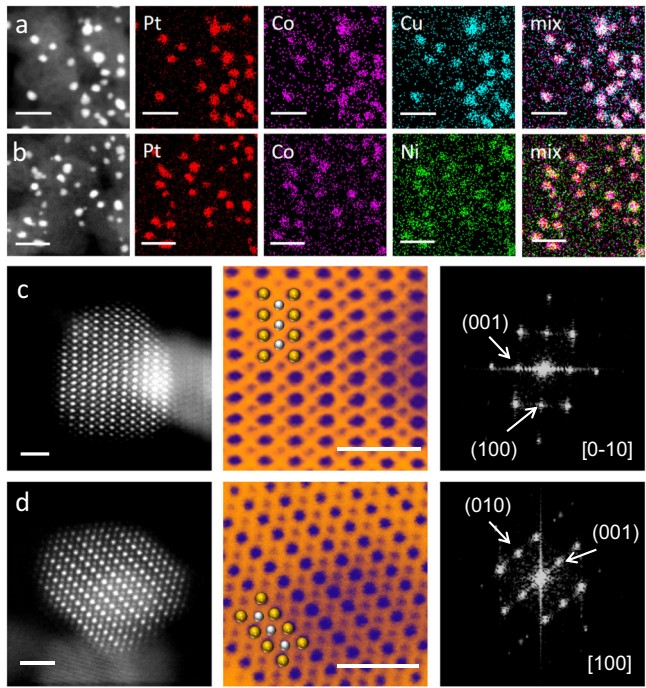

**Fig. 4 | Structure characterization.** EDS elemental mappings of the Pt$_2$CoCu (**a**) and Pt$_2$CoNi (**b**) catalysts, scale bar 10 nm. Atomic-resolution HAADF-STEM images and FFT patterns of Pt$_2$CoCu (**c**) and Pt$_2$CoNi (**d**) catalysts. Yellow balls represent Pt and silver balls represent CoCu (Ni). Scale bar: 1 nm.

relation, simultaneously showing a high SA (~3.5 mA cm$^{-2}$) and ECSA (~90 m$^2$ g$_{pt}^{-1}$), thus leading to a large MA of ~3.0 A mg$_{pt}^{-1}$. Moreover, after 30 K accelerated durability test (ADT) by cycling the potential between 0.6 and 0.95 V in RDE, the Pt$_2$CoCu/Pt$_2$CoNi catalysts showed a drop of 17.1% and 19.2% in the MA, along with a decrease of 10.2% and 22.2% in the SA (Fig. 5c, d and Supplementary Fig. S12). It has been well known that an IMCs@Pt core-shell structure would be formed in acid electrolytes and the Pt-shell could stabilize M against leaching to guarantee the durability under harsh voltage conditions[30,35].

We further evaluated the PEMFC performance of the Pt$_2$CoCu IMCs catalysts under practical H$_2$-air conditions. Prior to PEMFCs tests, the pristine IMCs catalysts were subjected to acid leaching and low-temperature H$_2$-annealing to form active and stable Pt-IMCs@Pt core-shell structures[8,11]. EDS elemental mapping indicated the successful formation of core-shell structure with a Pt-rich shell (Fig. 6a). Atomic resolution HAADF-STEM and corresponding intensity profiles clearly verified an L1$_0$ intermetallic core surrounded by a Pt shell with three atomic layers (Fig. 6b). Low Pt loadings of 0.056 mg$_{pt}$ cm$^{-2}$ and 0.020 mg$_{pt}$ cm$^{-2}$ were adopted for the Pt$_2$CoCu IMCs cathode and the commercial Pt/C anode, respectively. For comparison, commercial Pt/C was also tested as the cathode catalyst with a loading of 0.100 mg$_{pt}$ cm$^{-2}$. The low-Pt-loading Pt$_2$CoCu IMCs cathodes displayed high peak power densities of ~1.1 W cm$^{-2}$ at 150 kPa$_{abs}$/80 °C, which were comparable to that of Pt/C cathode at a nearly double Pt loading (Fig. 6c). The enhanced power performance of the Pt$_2$CoCu IMC catalyst in high-current-density region (>1.5 A cm$^{-2}$) was attributed to the advantage of high ECSA in minimizing mass-transport losses[36].

We finally evaluated the durability of the Pt$_2$CoCu and Pt/C cathodes from two metrics, voltage loss at 0.8 A cm$^{-2}$ and MA. After a

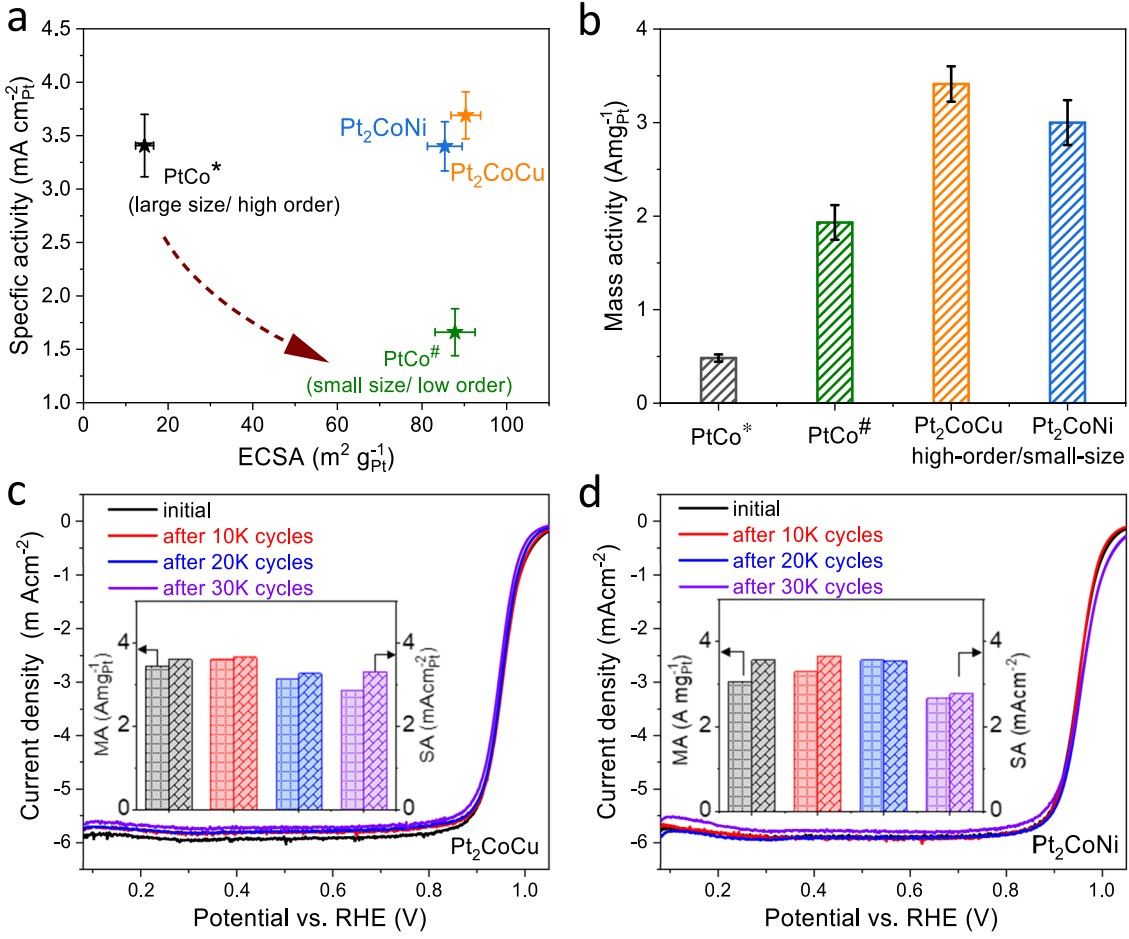

**Fig. 5 | RDE performance. a, b** Comparison of ECSA, SA, and MA of PtCo*, PtCo#, Pt$_2$CoCu, and Pt$_2$CoNi catalysts. ORR polarization curves and MA/SA loss of Pt$_2$CoCu (**c**) and Pt$_2$CoNi (**d**) catalysts after 30 K ADT.

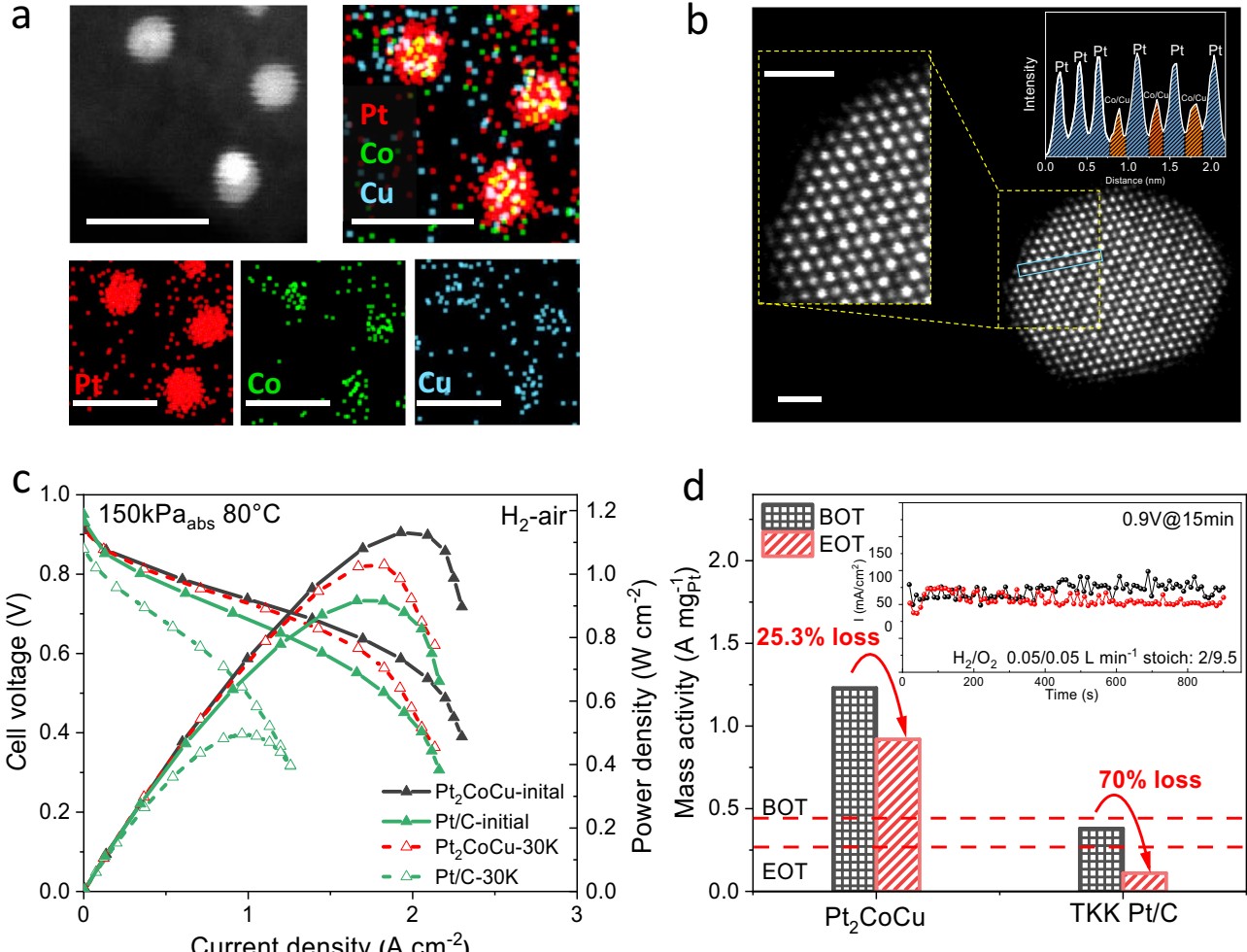

**Fig. 6 | Fuel cell performance. a** EDS elemental mappings of Pt$_2$CoCu@Pt core-shell structures. Scale bar: 10 nm. **b** Atomic resolution HAADF-STEM images and intensity profiles showing an intermetallic core with an thin Pt shell. Scale bar: 1 nm. **c** H$_2$/air polarization plots/power density plots before and after 30 K square wave

ADT. Test conditions: H$_2$/air 0.5/2 L min$^{-1}$, 80 °C, 100% relative humidity, 150 kPa$_{abs}$. **d** The loss of MA of EOT and BOT. Test conditions: H$_2$/O$_2$ 0.05/0.05 L min$^{-1}$, 80 °C, 100% relative humidity, 150 kPa$_{abs}$.

standard ADT protocol of a square wave potential cycling between 0.60 and 0.95 V (3-second hold at each voltage) suggested by US DOE (Supplementary Fig. S13), the current/power polarization plots at the beginning of test (BOT) and at the end of test (EOT) were obtained (Fig. 6c). The voltage loss for the Pt$_2$CoCu cathode at 0.8 A cm$^{-2}$ was 17 mV, which was much lower than the Pt/C cathode (-137 mV loss). Considering the instability of current in the high voltage region, DOE recommends multi-point test at 0.9 V$_{iR-free}$ for 15 min under a low cathode stoichiometry of 9.5, and the MA was calculated based on the average current density at the last 1 min (Fig. 6d). The Pt$_2$CoCu IMC catalyst retained 74.7% of its BOT-MA, which was much higher than Pt/C (30% MA retention) and exceeded the DOE target (less than 40% loss with an BOT MA of 0.44 A mg$_{Pt}^{-1}$).

In summary, aided by the machine-learning-accelerated computational screening, we established an experiment-theory-collaborative design strategy of small-sized and highly ordered IMC catalysts for fuel cells. Thanks to the well-defined ordered structures of IMC in atomic level, a better concordance between experiment and machine learning simulation can be guaranteed, which makes it feasible to quickly discover potential IMC combinations with high thermodynamic driving force for the disorder-to order transition from the enormous design space. Computationally, we found that alloying Cu/Ni promoted the formation of IMC due to the addtional stabilization energy introduced

by the Co-Cu/Co-Ni disordering. Experimental results demonstrated that the ternary Pt$_2$CoCu/Pt$_2$CoNi IMC catalysts could achieve the compatibility of small size (thus high ECSA) and high ordering degree (thus high SA), and finally achieved much improved mass-normalized performance in practical fuel cells.

## Methods

### DFT Calculation

All DFT calculations were performed using Vienna Ab Initio simulation packages (VASP)[37]. The interaction between the core and valence electrons was treated using projected-augmented wave (PAW) method[38]. A kinetic energy cutoff of 400 eV was used for the plane-wave basis set. The exchange-correlation energy was evaluated by the Perdew-Burke-Ernzerhof (PBE) functional at the generalized gradient approximation (GGA) level[39]. The energy tolerance of $5 \times 10^{-5}$ eV was used in electronic structure calculations, and the geometry optimization was performed until all force was less than 0.03 eVÅ$^{-1}$. The unit cell of PtCo in P4/mmm space group was downloaded from Material Project[40]. To construct the unit cell for Pt$_2$CoM, (M is the third element doped to PtCo), the unit cell of PtCo was repeated twice and a Co atom was replaced by an M atom. The lattice constants of all Pt$_2$CoM unit cells were re-optimized by DFT. To analyze the solubility and disorder-to-order transition energy of Pt$_2$CoM, a $(3 \times 3 \times 3)$ supercell was

constructed based on the DFT optimized Pt$_2$CoM unit cell. Atomic arrangement of initial Pt$_2$CoM supercell was shuffled to generate configurations with different ordering degrees. A $(3 \times 3 \times 3)$ Monkhorst-Pack k-point grid was utilized to sample the Brillouin zone for the supercell model[41]. To analyze the activity of Pt$_2$CoM, the close-packed Pt$_2$CoM (111) surfaces were simulated by a five-layer $(4 \times 4)$ slab model, in which two Pt$_2$CoM bottom layers were fixed and three Pt skin layers added on them were allowed to relax. Activity of ordered PtCo were evaluated using three Pt-skin layers with two ordered PtCo-core layers. The arrangement of Pt and Co in the ordered PtCo-core were shuffled to generate disordered PtCo slab models. A vacuum layer of 7.5 Å was added above and below the slab to avoid periodic interactions. A $(4 \times 4 \times 1)$ Monkhorst-Pack k-point grid was employed to sample the Brillouin zone for slab model. The spin polarization was included in the calculation of slab model for Fe, Co and Ni. The adsorption energy of O and OH were calculated by the following Equations:

$$\Delta E_O = E_{O^*} - E_* - (E_{H_2O} - E_{H_2}) \tag{1}$$

$$\Delta E_{OH} = E_{OH^*} - E_* - (E_{H_2O} - 1/2E_{H_2}) \tag{2}$$

where the $E_*$, $E_{O^*}$ and $E_{OH^*}$ are DFT calculated energies for clean surface, surface with O and OH, respectively. The $E_{H_2O}$ and $E_{H_2}$ are energies of H$_2$O and H$_2$ are −6.76 eV and −14.22 eV respectively, which are calculated by DFT using a H$_2$O and H$_2$ molecule placed on a 10 Å × 10 Å × 10 Å cubic cell model. The adsorption Gibbs free energy was defined by:

$$\Delta G = \Delta E + \Delta E_{water} + \Delta ZPE - T\Delta S \tag{3}$$

Where $\Delta E_{water}$ is the solvation correction, $\Delta ZPE$ is the zero point correction and $T\Delta S$ is the entropic correction, in this work we used correction values from the reference[32].

Short Range Order. The Warren-Cowley short range order is defined as Eq. (4)[42]:

$$\alpha^{X-Y} = 1 - \frac{P^{X|Y}}{c_X} \tag{4}$$

where X and Y denote two kinds of alloy elements, $c_X$ is the ratio of element X, $P^{X|Y}$ is the probability of finding X in the neighbor shell of Y. At a fixed $c_X$, a negative $\alpha^{X|Y}$ value suggests the preference for forming X-Y bond, which is characteristic of an ordered alloy; on the contrary, a positive $\alpha^{X-Y}$ value demonstrates the segregation of X and Y; and $\alpha^{X-Y}$ value close to 0 represents a random alloy configuration. Here the configurations with the lowest short range order value $\alpha^{Pt-(Co/M)} = -1/3$ are defined as order alloy, and configurations with $\alpha^{Pt-(Co/M)} = 0$ are regarded as random alloy.

**Solubility Test**
The initial 16 candidate elements include alkali metal Na; alkaline-earth metal Mg; 3d transition metals Sc, Ti, Mn, Fe, Ni, Cu; P-block elements Ga, Ge, In, Sn, Sb, Te, Pb, Bi. Given that the lowest and highest values of $\alpha^{Pt-(Co/M)}$ for the Pt$_2$CoM supercell model are -1/3 and 2/3 respectively, as depicted in Supplementary Fig. S6. 22 Pt$_2$CoM structures with different ordering degrees were generated (four structures for each $\alpha^{Pt-(Co/M)} = -1/6, 0, 1/6, 2/6$ and 3/6, one structure for $\alpha^{Pt-(Co/M)} = -2/6, 4/6$). Mente Carlo algorithm was applied to generate structures with the given $\alpha^{Pt-(Co/M)}$ value by manipulating the arrangement of Pt, Co and M in Pt$_2$CoM supercell model. Geometry optimizations were then conducted by DFT on these structures. Root mean square deviation (RMSD) was used to quantify the structural deformation during the geometry optimizations, RMSD is defined as:

$$\text{RMSD} = \frac{1}{n}\sqrt{\sum_{i=1}^{n}(X_i' - X_i)^2} \tag{5}$$

where n is the total number of atoms in the supercell model, $X_i'$ and $X_i$ are the positions of i-th atom before and after geometry optimization.

**Machine Learning**
Machine learning (ML) model was trained to replace DFT calculation to predict the energy for given alloy configuration. Extracting features from alloy structure as the input of ML model based on domain knowledge is of great importance for rationalizing the prediction model. Given that the stability of Pt$_2$CoM exhibits a strong correlation with the ordering degree (Supplementary Fig. S8), 15-dimensional numerical fingerprints reflecting the ordering degree of alloy configurations were designed and applied as the input of the ML model. The features consist of two parts. The Warren-Cowley short range order defined by three kinds of diatomic pairs (Pt-Co, Pt-M and Co-M) in three neighboring shells were used, which constituted the first nine dimensions in the feature. In addition, numbers of different diatomic bond pairs (Pt-Co, Pt-M, Co-M, Pt-Pt, Co-Co, M-M) were the other six dimensions for the feature. To avoid unbalanced weighting due to the different value ranges of each dimension in the feature, the original features were standardized to have a mean of 0 and a standard deviation of 1. Furthermore, to remove the redundant features and reduce calculation demands, principal component analysis (PCA) was carried out and only the principal components with a proportion of variance 99% were reserved. The DFT calculated relative energy of Pt$_2$CoM (energy reference to ordered Pt$_2$CoM supercell model) was used as the output for the ML model.

The Gaussian Process Regression (GPR) model was selected as the machine leaning model by taking advantage of its ability for uncertainty measurement[43]. Gaussian process regression is a non-parametric Bayesian inference regression technique. Unlike traditional regression method, Gaussian Process Regression gives a posterior distribution rather than exact value for the prediction target. Gaussian Process can be regarded as an infinite Gaussian distribution, and each final random variables set of Gaussian Process obeys a multivariate normal distribution, which can be denoted by:

$$f(X) \sim N(\mu(X), K(X)) \tag{6}$$

where $X = (x_1, x_2, \ldots, x_n)$ is a set of random variables, $f(X)$ is the target properties, $\mu(X)$ is mean of target properties, $K(X)$ is the covariance matrix. For the new datapoint $x^*$ needed to be predicted, the union set of $X$ and $x^*$ also obey following multivariate normal distribution:

$$\begin{bmatrix} f(X) \\ f(x^*) \end{bmatrix} \sim N\left( \begin{bmatrix} \mu(X) \\ \mu(x^*) \end{bmatrix}, \begin{bmatrix} K(X) & K(X,x^*)^T \\ K(X,x^*) & K(x^*,x^*) \end{bmatrix} \right) \tag{7}$$

The posterior distribution of $f(x^*)$ can be calculated by maximizing the likelihood function, and the results are expreseed as following:

$$f(x^*) \sim N(\mu(x^*), \sigma(x^*)) \tag{8}$$

$$\mu(x^*) = K(X,x^*)^T K(X)^{-1} f(X) \tag{9}$$

$$\sigma(x^*) = K(x^*,x^*) - K(X,x^*)^T K(X)^{-1} K(X,x^*) \tag{10}$$

The scikit-learn package was used for constructing GPR model[44]. Each term in the covariance matrix indicate the covariance between

two elements, which was defined by the radial-basis function (RBF):

$$k(x,x') = C^2 \exp\left(-\frac{(x-x')^2}{2l^2}\right) \qquad (11)$$

where $C$ and $l$ are scale factor and length factor, respectively. During the ML model training process, values of $C$ and $l$ were optimized to maximize the log-marginal-likelihood. The values range of $C^2$ and $l^2$ is limited to between $10^{-5}$ and $10^3$, and 50 independent optimizations were conducted to avoid local optimums.

## Active Learning
The workflow of active learning is shown in Supplementary Fig. S2. The uncertainty given by trained GPR model was used as the criterion to select samples in next generation. Due to the difficulty of exhausting the alloy configuration space, Monte Carlo simulation was performed to search the Pt$_2$CoM configuration with the high prediction uncertainty. Starting from a randomly initialized structure, Monte Carlo optimization controls the evolution of the structure along the high uncertainty direction through changing the atomic arrangement. When the uncertainty of the new structure is higher than that of the previous structure, the new structure was accepted; when the uncertainty of the new structure is lower than that of the previous structure, the new structure was accepted with a certain probability. In addition, to avoid getting trapped in local optimums, 20 simulations were performed and the first 10 structures with the highest uncertainty were extracted. The newly searched 10 structures were labeled by DFT and added into the pre-existing dataset. Then the ML model was retrained using the extended database. Such an active learning iteration was repeated 7 times. The searched Pt2CoCu configurations representing by their t-SNE in the active learning process was shown in Supplementary Fig. S3. The prediction accuracy of the ML model was measured using mean absolute error (MAE) and coefficient of determination (R2), where the 80% data was adopted as the training set and the remaining 20% data was the test set. The evolution of prediction accuracy in the active learning process are shown in Supplementary Figs. S4–5. The completed ML model was used to predict the energies of 300 ordered alloy structures and 300 random alloy structures.

## Materials and Chemicals
Carbon black (Black Pearls 2000, BP2000) was produced by America Cabot Corporation. Commercial Pt/C (TEC10E20E) was purchased from TANAKA. All others chemicals were commercially available from Sinopharm Chemical Reagent Co. Ltd., China, including chloroplatinic acid ($H_2PtCl_6\cdot6H_2O$), cobalt chloride hexahydrate ($CoCl_2\cdot6H_2O$), copper chloride dehydrate ($CuCl_2\cdot2H_2O$), cickel chloride hexahydrate ($NiCl_2\cdot6H_2O$), ethanol, and isopropanol. All the chemicals were used as received without further purification. DI water (18.2 M$\Omega$/cm) used in all experiments was prepared by passing through an ultra-pure purification system.

## Catalysts preparation
The Pt$_2$CoCu and Pt$_2$CoNi IMC catalysts were prepared with the BP2000 carbon black support by a conventional impregnation method that involved the wetness impregnation of metal salt and subsequent thermal reduction in 5% $H_2$/Ar. Taking the synthesis of Pt$_2$CoCu for an example, 50 mg BP2000, 20 mg $H_2PtCl_6\cdot6H_2O$, 6.4 mg $CoCl_2\cdot6H_2O$, and 3.9 mg $CuCl_2\cdot2H_2O$ was first mixed in a 100 mL round-bottom flask containing 40 mL DI water. After stirring overnight, the mixture was subjected to ultrasonic treatment for 2 h before drying by using a rotary evaporator. Finally, the dried powder was transferred to a tube furnace and thermally reduced at 1000 °C under flowing 5% $H_2$/Ar for 2 h.

## RDE Measurements
The Pt$_2$CoCu (Ni) IMC catalysts was assessed for their ORR activity using the RDE techniques using a CHI Electrochemical Station (Model 760E) in a three-electrode cell. The catalyst ink, comprising 4 mg of catalyst and 40 μL of Nafion in 2 mL of isopropanol, was prepared through sonication. Then, the ink was applied via drop-coating onto the working electrode, a 5.0 mm diameter glassy carbon disk, and left to air-dry at room temperature. Reference and counter electrodes were provided by saturated Hg/HgSO$_4$ and a platinum plate, respectively. All potentials in this study were referenced to the reversible hydrogen electrode (RHE) for each test. The catalyst underwent activation using cyclic voltammetry (scan rates of 250 mV s$^{-1}$ and potential ranges of 0.05–1.05 V vs. RHE) until achieving a stable curve. Linear sweep voltammetry (LSV) measurements were then conducted in O$_2$-saturated 0.1 M HClO$_4$ solution, sweeping the potential from 0.05 to 1.05 V at a rate of 10 mV/s (1600 rpm). Mass activity calculations involved capacitance-correction and IR-correction. For accelerated durability tests (ADTs), the potential was cycled between 0.6 and 0.95 V at a rate of 100 mV s$^{-1}$ at room temperature under N$_2$-saturated 0.1 M HClO$_4$ solution. The electrochemical active surface area (ECSA) was determined through a CO stripping test. CO stripping test involved bubbling CO into 0.1 M HClO$_4$ electrolyte, holding potential at 0.05 V for 10 min, followed by bubbling N$_2$ into the electrolyte for 30 min. Subsequently, a cyclic voltammetry curve was recorded by scanning from 0.05 V to 1.05 V at a rate of 50 mV s$^{-1}$.

## PEMFCs Tests
The catalysts first underwent acid treatment and subsequent annealing to create Pt-IMCs@Pt core-shell structures[8], which aimed to mitigate the poison effect of leached metal cation during the fuel cell operation. A uniform ink was prepared by dispersing catalysts in a solvent blend of n-propanol and water (1:1), incorporating Aquivion D72-25BS ionomer at an ionomer/carbon ratio of 0.8. The ink concentration was maintained at 3 mg$_{cat.}$ mL$^{-1}$. The catalyst-coated-membrane was created by applying an ultrasonic spray (ExactaCoat FC, Sono-Tek Corporation) on a GORE Nafion membrane (12 μm, 5 cm$^2$). A gas diffusion layer (GDL) was used Freudenberg (H24CX483, 235 μm) with a microporous layer. The membrane electrode assembly (MEA), incorporating two GDLs, two gaskets, and the prepared CCM, was assembled with a compression of 34%. The seven channel serpentine flow field was applied for the all single-cell tests (designed by Hubert Gasteiger and co-workers[45]), where the pressure drop between the inlet and outlet of the flow filed was less than 10 kPa.

The MEA was made with the Pt$_2$CoCu (0.056 mg$_{pt}$cm$^{-2}$) or benchmark Pt/C cathode (TEC10E20E, TANAKA, 0.1 mg$_{pt}$cm$^{-2}$) and the commercial Pt/C anode with a Pt loading 0.02 mg$_{pt}$cm$^{-2}$. Initially, calibration curves were established to quantify the correlation between the number of spray cycles and catalyst loading. This approach allows the attainment of a specific Pt loading by adjusting the number of spraying cycles. The reproducibility of the instrument ensures the accuracy of each spraying.

The mass activity (MA) of MEA was assessed at 0.9 V$_{iR-correct}$, 80 °C, 100% relative humidity, 150 kPa$_{abs}$, outlet H$_2$-O$_2$ at 0.05/0.05 L min$^{-1}$ flow rate (equivalent to the stoichiometry ratio of 2/9.5). A hold time of 15 min was implemented, and the mass activity calculated based on the average current during the last 1 min. The corresponding current was corrected for H$_2$ crossover. The H$_2$-air performance of single cell was conducted at 80 °C, 100% relative humidity, 150 kPa$_{abs}$, outlet H$_2$-air at 0.5/2 L min$^{-1}$ flow rate. For comparison, MEA made with Pt/C cathodes with a loading of 0.1 mg$_{pt}$ cm$^{-2}$ were also measured. According to the US DOE ADT protocol for PGM-based catalysts on carbon-based supports, the accelerated durability test (ADT) for the MEA involved applying a square wave voltage from 0.6 to 0.95 V, lasting 3 seconds at each voltage level. Each test was run up to 30,000

cycles at 80 °C, 100% RH, with $H_2/N_2$ flow 200/75 sccm for the anode and cathode, respectively.

## Characterization

XRD were performanced on a Japan Rigaku DMax-γA rotation anode x-ray diffractometer equipped with graphite monochromatized Cu-K radiation. HAADF-STEM images were produced on FEI Talos F200X operated at 200 kV. Atomic resolution HAADF-STEM images were produced on probe aberration-corrected JEM ARM200F (S) TEM operated at 200 kV. EDS mapping were used FEI Talos F200X equipped with Super X-EDS system.

## Data availability

All source data for DFT modeling and machine learning that were used in this study are available from the GitHub repository: https://github.com/ZhangLabTHU/PtCoM. Source data are provided with this paper.

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

## Acknowledgements

We acknowledge the funding support from the National Natural Science Foundation of China (Grants 22325903, 22221003, 22103047, 22305158, and 22071225), the National Key Research and Development Program of China (Grant 2018YFA0702001), the Plan for Anhui Major Provincial Science & Technology Project (Grants 202203a0520013 and 2021d05050006), the Joint Funds from Hefei National Synchrotron Radiation Laboratory (Grant KY2060000175), the China Postdoctoral Science Foundation (Grant 2022M722195 and 2023T160435) and USTC Research Funds of the Double First-Class Initiative (Grant YD2060002032).

## Author contributions

H.-W.L., L.Z. and P.Y. conceived and designed the project. L.Z., X.N. and K.C. carried out the DFT calculations and machine learning. P.Y., S.-B.L. and X.Z. synthesized and characterized the catalysts. M.Z. performed the HAADF-STEM characterization. P.Y. and S.-B.L. performed electrochemical test. P.Y., X.N., L.Z. and H.-W.L. co-wrote the paper. All the authors discussed the results and commented on the manuscript.

## Competing interests

The authors declare no competing interests.
