## [Peer Review File · Nature Communications]

Machine-learning-accelerated design of high-performance platinum intermetallic nanoparticle fuel cell catalystsREVIEWER COMMENTS

Reviewer #1 (Remarks to the Author):

This paper reports a machine-learning (ML) approach to intermetallic nanoparticle catalyst design, synthesis and test. ML-guided prediction gives them Pt₂CoM (M =Cu or Co) to be the best catalyst for ORR among 15 different M's the authors screened. Then the authors made Pt₂CoM nanoparticles and converted them into the desired intermetallic structure. As predicted, these intermetallic nanoparticles showed much enhanced catalysis for ORR in both H-cell and MEA tests. The work is of great interest for current searching of active and robust catalysts for fuel cell applications. The paper is generally well-written, but some important issues need to be clarified before the work can be considered for publication in this journal.

- 1) The authors started off PtCo and alloy PtCo with M to search for the best Pt₂CoM as catalysts. The authors did not mention what could happen if PtM are directly made and studied as catalysts for ORR. Intermetallic PtCo is known to be active and robust, and intermetallic PtCoNi has also been reported to show even higher activity towards ORR. The activity reported in this paper is similar to what has been reported. The authors should comment these more specifically, as these prior arts are more related to what they reported here in this paper.
- 2) It is not clear to me what makes Pt₂CoCu and Pt₂CoNi to be the optimum combination as the ORR catalyst. How can Cu or Ni make the PtCo more active? Is it possible to characterize CoCu or CuNi position in the alloy structure?
- 3) I do not understand Figure 1b. Is it true that MCo interacts more strongly with Pt than Co does? Is there any experimental evidence to support this calculated result?
- 4) My biggest concern is on the after-synthesis annealing step the authors used to obtain their intermetallic nanoparticles. Although they commented on nanoparticle sintering issues in previous high temperature annealing steps, they used even higher temperature (1000C) to anneal their nanoparticle sample for the formation of intermetallic structure. Such a high temperature annealing in the presence of carbon and M is known to form M-C, which has also been reported to be active for ORR. Some control experiments on the annealed CoCu, and/or CuNi + carbon samples are necessary to support their main conclusion in this paper.
- 5) In the ORR and MEA tests, the authors just commented on the stability of their catalyst with some very vague words, "a negligible shift of polarization curve". There is no data to support if the intermetallic structure is both structurally and compositionally stable during the ORR or MEA tests. The authors only cited a previous work to assume their catalyst has a core/shell structure after the tests – "It has been well known that an IMCs@Pt core-shell structure would be formed in acid electrolytes and the Pt-shell could effectively stabilize M against leaching to guarantee the durability under harsh voltage conditions 26, 29". This is not the proof that their nanoparticles have the same structure.

Reviewer #2 (Remarks to the Author):

This paper presents a machine learning study to screen catalyst for PEM fuel cells. They identified two potential ternary candidates, which were verified by their experiment. The ordered structures were characterized using various imaging methods. The fuel cell performance using the catalyst seems outstanding, superior to state-of-the-art in terms of both Pt loading and performance. The paper is well prepared, and the results are exciting.

The method can be beneficial. After the below comments are addressed, I recommend for publication in the Journal:

- 1.) For machine learning, it usually requires big data during training to obtain reliable outcomes. Can they authors clarify this size of training data in their work?
- 2.) “corresponding non-noble metal salts on a commercial carbon support,”: the authors need to give details regarding commercial carbon support, materials and band.
- 3.) Can the authors look at this paper: “Hu, B., Yuan, J., Zhang, J., Shu, Q., Guan, D., Yang, G., ... & Shao, Z. (2021). High activity and durability of a Pt–Cu–Co ternary alloy electrocatalyst and its large-scale preparation for practical proton exchange membrane fuel cells. *Composites Part B: Engineering*, 222, 109082.” Which also shows outstanding performance of PtCoCu?
- 4.) It is very impressive that the machine learning tool can identify the catalyst materials that perform superiorly, both performance and durability. The authors may need to find similar literature work (maybe in general field) that supports this kind of identification.
- 5.) Can the authors look at this paper: “Wang, Y., Seo, B., Wang, B., Zamel, N., Jiao, K., & Adroher, X. C. (2020). Fundamentals, materials, and machine learning of polymer electrolyte membrane fuel cell technology. *Energy and AI*, 1, 100014.” Which summarizes a few general machine learning tools for material/chemistry screening?
- 6.) Can the authors provide details regarding their fuel cell configuration and operating condition such as GDL, CL thickness, RH,....?

Reviewer #3 (Remarks to the Author):

In this contribution, authors describe machine learning application to down select fuel cell catalyst in the Pt₂CoM family. Based on the active learning, informed by DFT, the authors prepared Pt₂CoCu and Pt₂CoNi catalysts and performed experimental analysis. Overall, the paper is publishable in JPCC and Chemistry of Materials. This reviewer doesn't recognize this manuscript is offering any new significant insights of PEMF catalysts or machine learning to satisfy nature communication criteria. Some of the general comments are provided below.

The active learning (based on GPR) and DFT (PBE calculations) associated with the paper are standard practices used in ML and literature. No new computational or chemical insights were provided. What's new here ? Does ML is needed to identify M= Ni or Copper?

Importantly, the identification of Cu or Nickel is not surprising (there are many in the literature, see <https://doi.org/10.1039/D0RA05468B> , & Sapkota et al, 2022).

The advantage or need and configurational materials space of the catalysts were not described. No quantitative metrics of how machine learning accelerated the materials discovery is explained.

No GitHub details were provided.

There are statements such as PtCoCu or PtCoNi show high ORR activity. No explanations or valid hypothesis were mentioned. Detailed DFT studies are needed to show the binding sites and demonstrate reaction pathways.

Overall, this reviewer does not recommend this paper for Nature Communications

Reviewer #1 (Remarks to the Author):

This paper reports a machine-learning (ML) approach to intermetallic nanoparticle catalyst design, synthesis and test. ML-guided prediction gives them Pt₂CoM (M =Cu or Co) to be the best catalyst for ORR among 15 different M's the authors screened. Then the authors made Pt₂CoM nanoparticles and converted them into the desired intermetallic structure. As predicted, these intermetallic nanoparticles showed much enhanced catalysis for ORR in both H-cell and MEA tests. The work is of great interest for current searching of active and robust catalysts for fuel cell applications. The paper is generally well-written, but some important issues need to be clarified before the work can be considered for publication in this journal.

Response: We sincerely appreciate the reviewer for the positive comments.

1) The authors started off PtCo and alloy PtCo with M to search for the best Pt₂CoM as catalysts. The authors did not mention what could happen if PtM are directly made and studied as catalysts for ORR. Intermetallic PtCo is known to be active and robust, and intermetallic PtCoNi has also been reported to show even higher activity towards ORR. The activity reported in this paper is similar to what has been reported. The authors should comment these more specifically, as these prior arts are more related to what they reported here in this paper.

Response: Many thanks for the reviewer's comments on this issue.

PtCo represents the most promising low-Pt catalyst for practical PEMFCs applications. For example, Toyota has launched several fuel-cell vehicles based on the binary PtCo alloy catalysts. Therefore, in this work, we performed our theoretical and experimental studies by focusing on the binary PtCo and ternary PtCoM systems and we did not extend our studies to other binary systems, such as PtCu or PtNi.

We agree with the reviewer that binary PtCo and even ternary PtCoM (M=Cu, Ni, etc) ORR catalysts have been widely reported. Despite these prior works match our computational screening results, there are notable differences that distinguish our research from reported ones. We focus more on the formation process of IMC, aiming to maintain the intrinsic high strain levels of PtCo without sacrificing the active surface area. We therefore resorted to machine learning to expedite the screening process for

identifying combinations with high thermodynamic driving forces. This enables us to achieve a balance between intrinsic activity and ECSA, ultimately leading to high mass activity that is meaningful for practical applications. As suggested by the reviewer, we have already cited and discussed the related works reported by others (PtCoNi, *J. Am. Chem. Soc.* 2020, 142, 45; PtCoCu, *Small*, 2023, 19, 2300112).

[Revision to manuscript] (Page1):

Carbon supported platinum-based intermetallic alloys are promising candidates as low-platinum oxygen reduction reaction electrocatalysts for proton-exchange-membrane fuel cells. Intermetallic PtCo is known to be one of the most promising intermetallic fuel cell catalyst.

[Revision to manuscript] (Page7):

..., and thus were finally selected for the experimental validation. *There has been prior research reporting PtCoNi and PtCoCu ternary system as excellent ORR catalyst due to their near-optimum strain levels for higher ORR activity^{33,34}, aligning well with our computational screening results. More importantly, our study provides a novel design perspective that the introduction of Cu/Ni leverages the thermodynamic driving force for the disordered-to-order transition, resulting in a more favorable tradeoff between specific activity (SA) and electrochemically active surface area (ECSA).*

2) It is not clear to me what makes Pt₂CoCu and Pt₂CoNi to be the optimum combination as the ORR catalyst. How can Cu or Ni make the PtCo more active? Is it possible to characterize CoCu or CuNi position in the alloy structure?

Response: We would like to thank the reviewer for the above constructive suggestions.

(a) “How can Cu or Ni make the PtCo more active?”

PtCo demonstrates remarkable efficiency as an ORR catalyst, however the formation of highly ordered intermetallic alloys often necessitates elevated annealing temperatures and inevitably results in particle sintering issues, leading to a reduction in the effective electrochemical surface area. Our work aims to enhance the ordering of Pt-Co alloy by improving thermodynamic driving forces. Computationally, we show that the screened element Cu and Ni can effectively facilitate the Pt-Co ordering. Thus,

a catalyst that combines high intrinsic activity (high SA) and small size (high ECSA) can be achieved. As shown in the Figure R1, the improved tradeoff between ordering (higher activity) and size (larger ECSA) leads to the optimized ORR performance.

Fig. R1. The introduction of Cu/Ni break the trade-off between intrinsic activity and active surface area in the synthesis of PtCo.

Fig. R2. ORR activity volcano plot as a function of OH and O adsorption energy relative to Pt of four ordered Pt₂CoM, fully-ordered PtCo and five randomly-ordered PtCo.

Based on experimental and theoretical results, **the introduction of Cu/Ni does not necessarily enhance the intrinsic activity of PtCo.** Instead, it plays an essential role in augmenting the thermodynamic driving force for the disordered-to-order transition process, consequently promoting the production of highly-ordered intermetallic

compounds (IMCs). The calculated adsorption energies of OH* and O* were plotted as the activity descriptor to evaluate the ORR activity of the Pt₂CoM (M= Ga, Ni, Cu, and Ti) using Pt-shell slab models with ordered Pt₂CoM-core (Fig. R2). For comparison, we also marked the calculated activity of fully ordered and randomly ordered PtCo with various arrangements of subsurface PtCo core. Generally, fully ordered PtCo showed higher activity than randomly ordered one, and ternary Pt₂CoCu and Pt₂CoNi show comparable ORR activity with fully ordered PtCo. Experimentally, the RDE results also demonstrated that the specific activity of Pt₂CoCu and Pt₂CoNi is close to that of the highly-ordered PtCo (Fig. R1).

According to the reviewers' suggestions, we have added the following sentences in the main text to highlight the above discussion:

[Revision to manuscript] (Page7):

For ternary alloys, Pt₂CoCu and Pt₂CoNi show comparable ORR activity with fully ordered PtCo, and were finally selected for the experimental validation. ...More importantly, our study provides a novel design perspective that the introduction of Cu/Ni leverages the thermodynamic driving force for the disordered-to-order transition, resulting in a more favorable tradeoff between specific activity (SA) and electrochemically active surface area (ECSA).

(b) "Is it possible to characterize CoCu or CuNi position in the alloy structure?"

It was difficult to distinguish on the basis of the contrast due to the similar atomic radius and similar atomic weight. A previously published work (*J. Am. Chem. Soc.* 2020, 142, 45) has also tried to identify this issue, but failed. However, our theoretical prediction found the positions of CoCu or CuNi have a significant impact on the energy of ternary systems. The disordering of Co-Cu or Cu-Ni can provide additional thermodynamic stabilization energy (Figure R3), suggesting that the Co with Cu/Ni tend to have random site occupation in the ternary Pt₂CoCu/Pt₂CoNi alloy structure.

Fig. R3. The effect of CoCu (a) or CuNi (b) position for the energy of ternary systems.

[Revision to manuscript] (Page 8-9) :

..., we could observe an alternating bright and dark stacking of Pt and non-noble metal columns. *Our theoretical predictions suggest that Co with Cu or Ni tends to exhibit random site occupation. However, distinguishing the Co/Cu or Co/Ni position in the alloy structure is extremely challenging due to their similar atomic radius (J. Am. Chem. Soc. 2020, 142, 45).*

3) I do not understand Figure 1b. Is it true that MCo interacts more strongly with Pt than Co does? Is there any experimental evidence to support this calculated result?

Response: Many thanks for the reviewer's comments on this issue. The original Figure 1b was a schematic plot, illustrating the presence of Pt₂CoM combination with potentially higher thermodynamic driving force of disordered-to-ordered transition. The ordering energy E_{ordering} measures the thermodynamic driving force for the disorder-to-order transition, which is defined as the energy difference between ordered configurations (SRO of Pt-Co/M: $\alpha^{\text{Pt-(Co/M)}} = -1/3$) and randomly mixed configurations ($\alpha^{\text{Pt-(Co/M)}} = 0$). As shown in Fig 2C, various metals exhibit distinct effects on the ordering energy E_{ordering} . Among them, the screened metals, Cu and Ni, prove to be effective in leveraging the ordering energy, and consequently were selected for experimental validation.

The correlation between the ordering energy trend and the Pt-Co-M pair interactions is an interesting topic, but falls beyond the scope of the current study. We are working on this issue and will present our findings in future work

Taking into account the reviewer's feedback, we have revised this schematic illustration to more accurately represent the catalyst design guideline of this work, specifically emphasizing the incorporation of a third metal to enhance the thermodynamic driving forces. (Figure R4).

Fig. R4. Schematic illustration showing that the thermodynamic driving force of disordered-to-ordered transition could be enhanced by forming ternary IMC structure.

The dilemma of the PtCo synthesis often revolves around a lower ordering degree, which potentially inhibits its specific activity and durability. Thus, high temperature has to be used to promote the IMC nucleation with the expense of losing ECSA (e.g. high-ordered/ large-sized PtCo^{*}). Therefore, we aim at the *de novo* design of the element composition to increase the thermodynamic driving force for the disordered-to-order transition and thus promote the ordering degree. By comparing similar particle size of PtCo[#] and PtCoCu/Ni, without sacrificing ECSA, it is observed that PtCoCu/Ni (>50% ordering degree) show a higher ordering degree than that of PtCo[#] (~5%).

Fig R5. The comparison of ordering degree between PtCoCu/Ni and PtCo under the similar size.

[Revision to manuscript] (Fig. 1b) :

4) My biggest concern is on the after-synthesis annealing step the authors used to obtain their intermetallic nanoparticles. Although they commented on nanoparticle sintering issues in previous high temperature annealing steps, they used even higher temperature (1000C) to anneal their nanoparticle sample for the formation of intermetallic structure. Such a high temperature annealing in the presence of carbon and M is known to form M-C, which has also been reported to be active for ORR. Some control experiments on the annealed CoCu, and/or CuNi + carbon samples are necessary to support their main conclusion in this paper.

Response: Many thanks for the reviewer's comments on this issue. As suggested by the reviewer, we synthesized the annealed CoCu/C and CuNi/C catalysts to

performance RDE test. Compared to the Pt-based intermetallic catalyst, the ORR activity of the CoCu/C or CoNi/C catalyst before and after acid etching could be negligible (Figure R6).

Fig R6. XRD and RDE test of CoCu/C and CuNi/C catalysts.

5) In the ORR and MEA tests, the authors just commented on the stability of their catalyst with some very vague words, “a negligible shift of polarization curve”. There is no data to support if the intermetallic structure is both structurally and compositionally stable during the ORR or MEA tests. The authors only cited a previous work to assume their catalyst has a core/shell structure after the tests – “It has been well known that an IMCs@Pt core-shell structure would be formed in acid electrolytes and the Pt-shell could effectively stabilize M against leaching to guarantee the durability under harsh voltage conditions 26, 29”. This is not the proof that their nanoparticles have the same structure.

Response: Many thanks for the reviewer’s comments on this issue. We have revised these vague expressions. During the ADT test in RDE, the Pt₂CoCu/Pt₂CoNi catalysts showed a drop of 17.1% and 19.2% in the MA, along with a decrease of 10.2% and 22.2% in the SA. For the ADT test in MEA, the Pt₂CoCu catalysts showed a 25.3% loss

of MA. The MEA performance degradation was due to the non-noble metal dissolution and particle sintering. In the XRD analysis of Pt₂CoCu-MEA after ADT (Fig. R7), we observed that the super-lattice peaks became weaker, indicating the dissolution of CoCu and consequent degradation in performance.

Fig R7. XRD pattern of Pt₂CoCu-MEA after ADT.

“This is not the proof that their nanoparticles have the same structure.”

As suggested by the reviewer, we further performed energy dispersive spectroscopy (EDS) elemental mapping and aberration-corrected HAADF-STEM to verify IMCs@Pt core-shell structure. EDS elemental mapping indicated the successful formation of core-shell structure with a Pt-rich shell (Fig. R8). Atomic resolution HAADF-STEM and corresponding intensity profiles clearly verified an L1₀ intermetallic core surrounded by three atomic layers of a Pt shell (Fig. R9).

Fig. R8. EDS elemental mappings of Pt₂CoCu@Pt core-shell structures.

Fig. R9. Atomic resolution HAADF-STEM images of Pt₂CoCu@Pt core-shell structures.

[Revision to manuscript]:

Fig. 6 | Fuel cell performance. (a) EDS elemental mappings of Pt₂CoCu@Pt core-shell structures. (b) Atomic resolution HAADF-STEM images and intensity profiles showing an intermetallic core with a thin Pt shell. (c) H₂/air polarization plots/power density plots before and after 30K square wave ADT. (d) The loss of MA of EOT and BOT.

Page 10:

..., thus leading to a large MA of $\sim 3.0 \text{ A mg}_{\text{Pt}}^{-1}$. Moreover, after 30K accelerated durability test (ADT) by cycling the potential between 0.6 and 0.95 V in RDE, the Pt₂CoCu/Pt₂CoNi catalysts showed a slight drop of 17.1% and 19.2% in the MA, along

with a decrease of 10.2% and 22.2% in the SA (Fig. 5c,d and Fig. S12).

Page 10:

Prior to PEMFCs tests, the pristine IMCs catalysts were subjected to acid leaching and low-temperature H₂-annealing to form active and stable Pt-IMCs@Pt core-shell structures^{8,11}. *EDS elemental mapping indicated the successful formation of core-shell structure with a Pt-rich shell (Fig. 6a). Atomic resolution HAADF-STEM and corresponding intensity profiles clearly verified an L1₀ intermetallic core surrounded by a Pt shell with three atomic layers (Fig. 6b).*

Reviewer #2 (Remarks to the Author):

This paper presents a machine learning study to screen catalyst for PEM fuel cells. They identified two potential ternary candidates, which were verified by their experiment. The ordered structures were characterized using various imaging methods. The fuel cell performance using the catalyst seems outstanding, superior to state-of-the-art in terms of both Pt loading and performance. The paper is well prepared, and the results are exciting. The method can be beneficial. After the below comments are addressed, I recommend for publication in the Journal:

Response: We sincerely appreciate the reviewer for the positive comments.

1) For machine learning, it usually requires big data during training to obtain reliable outcomes. Can they authors clarify this size of training data in their work?

Response: Many thanks for the reviewer's comments on this issue. The sizes of each DFT-calculated configuration are summarized in Table R1. As the review said, for machine learning, a significant amount of data is typically required to ensure reliable results. To reduce the data requirements in DFT, active learning is employed in our work. Active learning begins with a small initial dataset and strategically incorporates new data points into the dataset and retrain the model iteratively, which significantly reduces the computation intensity. This approach effectively mitigates issues of uneven or insufficient data points, consequently reducing data size demands.

In Table 2R, we compiled a list of studies employed machine learning method in predicting alloy configuration energies. While these studies vary in the problems studied, dataset construction, and selected machine learning models, it is evident that, for the specific task of predicting PtCoM configuration energies in this study, the employed method achieves a comparable level of accuracy while requiring less data than reported in the literature.

Table R1. The size of DFT calculated configurations for the different Pt₂CoM. The minor difference in data size is attributed to the removal of duplicate configurations.

IMC	Size
PtCo	109
Pt ₂ CoCu	120
Pt ₂ CoNi	116
Pt ₂ CoMn	106
Pt ₂ CoFe	116
Pt ₂ CoGa	107
Pt ₂ CoTi	116

Table R2. A list of studies employed machine learning method in predicting alloy configuration energies.

System	Datasize	MAE (meV/Atom)	ML model	Ref
PtCo(M)	107~120	2.7~9.6	GPR	Our Work
CdPdAu	5278	1.8	Behler– Parrinello neural network	J. Phys. Chem. C 2022, 126, 4, 1800–1808
Co ₃ W	1000	3.8	Schnet	Adv. Funct. Mater. 2022, 32, 2208418
RuAg	1106	4~6	Bayesian linear regression	J. Phys. Chem. Lett. 2017, 8, 17, 4279–4283
MoNbTaW	5564	3.76	Machine learning potential	Acta Mater. 2023: 119041.
Li _x TiS ₂	1000	6	Neural network	npj Comput. Mater. 2018, 4(1): 56.

Co ₃ Al(M)	180	16	Support Vector Regression	Comput. Mater. Sci. 2021, 200: 110787.
-----	----	------------------------------	---

[Revision to Supplemental information]: (Page15):

... , *Supplementary Table 2. The size of DFT calculated configurations for the different Pt₂CoM.*

2) “corresponding non-noble metal salts on a commercial carbon support,”: the authors need to give details regarding commercial carbon support, materials and band.

Response: Many thanks for the reviewer’s comments on this issue. Material details are presented in the experimental section. Commercial carbon support was used the Black Pearls 2000 (BP2000). Others precursor salts included chloroplatinic acid (H₂PtCl₆·6H₂O), cobalt chloride hexahydrate (CoCl₂·6H₂O), copper chloride dehydrate (CuCl₂·2H₂O), and nickel chloride hexahydrate (NiCl₂·6H₂O).

[Revision to manuscript] (Page7):

The Pt₂CoCu and Pt₂CoNi catalysts were synthesized by the wet-impregnation of H₂PtCl₆ and corresponding non-noble metal salts on a carbon support Black Pearl 2000, followed by a high-temperature annealing at 1000 °C.

3) Can the authors look at this paper: “Hu, B., Yuan, J., Zhang, J., Shu, Q., Guan, D., Yang, G. ... & Shao, Z. (2021). High activity and durability of a Pt–Cu–Co ternary alloy electrocatalyst and its large-scale preparation for practical proton exchange membrane fuel cells. Composites Part B: Engineering, 222, 109082.” Which also shows outstanding performance of PtCoCu?

Response: We would like to thank the reviewer for bringing this remarkable work to our attention. The author reported PtCoCu as an efficient and durable fuel cell oxygen reduction catalyst, mainly attributing the improved performance to significant compressive strain. Our work offers a novel perspective by optimizing the chemical

ordering. According to the reviewer's suggestion, we have cited this paper in the revised manuscript.

4) It is very impressive that the machine learning tool can identify the catalyst materials that perform superiorly, both performance and durability. The authors may need to find similar literature work (maybe in general field) that supports this kind of identification.

Response: We appreciate the reviewer for the above kind suggestions. We added the following citations on machine learning for material discovery.

[Revision to manuscript] (Page3) :

Recently, machine learning methods have demonstrated significant potential in accelerating material discovery by efficiently navigating design spaces and predicting properties, thereby substantially reducing the cost of identifying and optimizing novel materials²⁶⁻²⁹.

5.) Can the authors look at this paper: “Wang, Y., Seo, B., Wang, B., Zamel, N., Jiao, K., & Adroher, X. C. (2020). Fundamentals, materials, and machine learning of polymer electrolyte membrane fuel cell technology. Energy and AI, 1, 100014.” Which summarizes a few general machine learning tools for material/chemistry screening?

Response: We are grateful to the reviewer for introducing this outstanding work to our notice. As per the reviewer's recommendation, we have carefully added this paper to our citation.

[Revision to manuscript] (Page3) :

Recently, machine learning methods have demonstrated significant potential in accelerating material discovery by efficiently navigating design spaces and predicting properties, thereby substantially reducing the cost of identifying and optimizing novel materials.²⁶⁻²⁹.

6.) Can the authors provide details regarding their fuel cell configuration and operating condition such as GDL, CL thickness, RH,....?

Response: Many thanks for the reviewer's comments on this issue. The fuel cell

configuration and operating condition are presented in the experimental section. A gas diffusion layer (GDL) was used Freudenberg (H24CX483, 235 μm) including a microporous layer. The catalysts layer (CL) thickness of the cathode and anode are about 5 μm and 2 μm . Two GDLs, two gaskets, and the prepared CCM constitute the membrane electrode assembly (MEA) with a 34% compression. Single fuel cell test station was Scribner 850e. The seven channel serpentine flow field was applied for the all single-cell tests (designed by Hubert Gasteiger and co-workers), where the pressure drop between the inlet and outlet of the flow field was less than 10 kPa. The H₂-air performance of single cell was conducted at 80 °C, 100% relative humidity, 150 kPa_{abs}, outlet H₂-air at 0.5/2 L min⁻¹ flow rate.

[Revision to manuscript] (Page17) :

The concentration of the ink was controlled to be 3 mg_{cat.} mL⁻¹. The catalyst-coated-membrane (CCM) was made on GORE Nafion membrane (12 μm , 5 cm²) using an ultrasonic spray (ExactaCoat FC, Sono-Tek Corporation). A gas diffusion layer (GDL) was used Freudenberg (H24CX483, 235 μm) including a microporous layer. Two GDLs, two gaskets, and the prepared CCM constitute the membrane electrode assembly (MEA) with a 34% compression.

Reviewer #3 (Remarks to the Author):

In this contribution, authors describe machine learning application to down select fuel cell catalyst in the Pt₂CoM family. Based on the active learning, informed by DFT, the authors prepared Pt₂CoCu and Pt₂CoNi catalysts and performed experimental analysis. Overall, the paper is publishable in JPCC and Chemistry of Materials. This reviewer doesn't recognize this manuscript is offering any new significant insights of PEMF catalysts or machine learning to satisfy nature communication criteria. Some of the general comments are provided below.

Response: We are sorry that the novelty of our work may have not been fully conveyed to the reviewer. Here, we would like to reemphasize the new insights of our work offers, specifically the correlation between chemical ordering and performance. We hope this clarification will help highlight its significance. Detailed point-to-point responses are provided below.

1. The active learning (based on GPR) and DFT (PBE calculations) associated with the paper are standard practices used in ML and literature. No new computational or chemical insights were provided. What's new here? Does ML is needed to identify M=Ni or Copper?

"No new computational or chemical insights were provided. What's new here?"

Response: Thanks for the candid feedback from the reviewer. Our primary focus is on enhancing the ordering of Pt-Co alloys by introducing a third element, thereby reducing the annealing temperature to achieve a highly ordered alloy. Computational screening reveals that the addition of Cu and Ni effectively promotes ordering in Pt-Co without compromising the alloy's oxygen reduction reaction (ORR) activity. Utilizing machine learning techniques, we statistically correlate chemical ordering and relative stability, enabling a quantitative assessment of the thermodynamic driving force behind intermetallic alloy formation. Furthermore, our findings highlight that the ordering of Pt-Co contributes to alloy stability, and the introduction of disorder between Co-Cu/Ni provides additional stabilization energy. All of the aforementioned insights are novel findings resulting from our computational work.

To the best of our knowledge, our work is among the earliest endeavors, if not the first attempt, to statistically quantify the impact of chemical ordering on thermodynamic stability with the aid of machine learning method. Additionally, the distinct effects of Pt-Co/M ordering and Co-Cu/Ni ordering are reported for the first time. The well-defined ordered structures of intermetallic compounds (IMC) at the atomic level ensure a more accurate alignment between experimental and machine learning simulations. This alignment facilitates the identification of potential IMC combinations with a high thermodynamic driving force for the disorder-to-order transition within the vast design space.

Based on the reviewer's feedback, we have made the following changes to the main text to better convey our novelty.

[Revision to manuscript]:

1. Abstract (Page1):

...we discover potential Pt_2CoM (M represents base metal element) combinations with high thermodynamic driving force for the disorder-to-order transition. With the aid of machine learning method, we are able to statistically quantify the impact of chemical ordering on thermodynamic stability. In particular, the alloying of Cu or Ni, inducing Co-Cu or Co-Ni disorder, provides additional stabilization energy, facilitating the ordering process and an improved tradeoff between electrochemically active surface area (ECSA) and specific activity (SA). Guided by the theoretical prediction, the small sized and highly ordered Pt_2CoCu and Pt_2CoNi catalysts are experimentally prepared, showing a large ECSA of $\sim 90\text{ m}^2\text{ g}_{Pt}^{-1}$ and a high SA of $\sim 3.5\text{ mA cm}^{-2}$.

2. Conclusion (Page 12):

..., which makes it feasible to quickly discover potential IMC combinations with high thermodynamic driving force for the disorder-to order transition from the enormous design space. Computationally, we found that alloying Cu or Ni promotes the formation

of IMC due to the additional stabilization energy introduced by the Co-Cu or Co-Ni disordering.

“Does ML is needed to identify M= Ni or Copper?”

Response: We thank the reviewer for raising this question, and our answer is affirmative. Machine learning (ML) plays a crucial role in computing the ordering energy and comprehending the stabilization factor for Ni and Cu doping. Quantitatively evaluating the thermodynamic driving force of Pt₂CoM transitions from disorder to order requires evaluating a significant number of configurations at different chemical orderings in Pt₂CoM. Given the diversity in the atomic arrangement of Pt, Co, and M elements, this constitutes an extensive space, making it impractical to conduct density functional theory (DFT) calculations for all configurations. To address this challenge, machine learning method was employed, utilizing a limited amount of DFT data (~100 per Pt₂CoM system) as the training dataset, to predict the energies of additional Pt₂CoM configurations with various chemical orderings. This approach substantially reduces both DFT computational costs and time expenditures.

As our super bulk model of Pt₂CoM contains 108 atoms (54 Pt atoms, 27 Co atoms and 27 M atoms), if we neglect the symmetry, the total number of isomers for this 108-atom Pt₂CoM model can be a rather extensive number. We employ active learning in a batch-wise manner to choose just over 100 representative configurations for DFT calculations. This allows for a comprehensive and unbiased representation of the entire configuration space.

In particular, one DFT geometry optimization task for Pt₂CoM bulk consumes approximately 2 hours when executed on two 2.1 GHz 28-core CPU nodes. In contrast, training a GPR model and utilizing it for energy prediction can be accomplished within just one minute on a personal computer, this time is negligible when compared with DFT computations, the majority of the time spent in machine learning is on generating the DFT training dataset. Specifically, as shown in Figure R10, through 120 DFT data

points, only a rough correlation between structural stability and chemical ordering can be discerned. However, by utilizing machine learning to quickly analyze 190 varying ordering degrees (20 configurations for each data point, in total 3800 configurations), a clearer structure-performance relationship as a function of Pt-Co/M and Co-M ordering can be established, saving nearly 32 times. It is essential to note that the time savings achieved through machine learning become even more significant when applied to a larger size of supercell or a greater number of structures.

Fig. R10. DFT calculates (left) and ML predicted relative energies of Pt_2CoCu configurations as a function of Pt-Co/Cu SRO $\alpha_1^{\text{Pt}-(\text{Co}/\text{Cu})}$ and Co-Cu SRO $\alpha_1^{\text{Co}-\text{Cu}}$. The number of data points of DFT calculated configuration and ML predicted are 120 (1 configurations per data point) and 3800 (average of 20 configurations for each data point), respectively.

[Revision to manuscript]

1. Page 3 :

..., which would significantly limit the nucleation rate of IMC phase.

Recently, machine learning methods have demonstrated significant potential in accelerating material discovery by efficiently navigating design spaces and predicting properties, thereby substantially reducing the cost of identifying and optimizing novel materials.²⁶⁻²⁹

2. Page 6 :

A higher SRO value represents a higher disordered degree. Triangles and circles represent data points computed by DFT (1 configurations per data point) and ML

prediction model (average of 20 configurations for each data point), respectively.

2. Page5 :

The trained machine learning model was applied to predict the formation energies of 300 ordered ($\alpha^{\text{Pt}-(\text{Co}/\text{M})} = -1/3$) and 300 random ($\alpha^{\text{Pt}-(\text{Co}/\text{M})} = 0$) configurations, as well as 3800 structures with varying ordering degrees (Fig. S6).

3. Page5 :

..., but the introduction of Mn and Fe would significantly suppress the thermodynamic driving force for the disorder-to-order transition of Pt₂CoM (Fig. 2c and Table S1). In the realm of computational efficiency, training and predicting with machine learning models take negligible time compared to DFT calculations, leading to a significant reduction in the time required to establish the correlation between chemical ordering and stability (Fig. S9).

2. Importantly, the identification of Cu or Nickel is not surprising (there are many in the literature, see <https://doi.org/10.1039/D0RA05468B> , & Sapkota et al, 2022).

Response: We thank the reviewer for suggesting these literatures. As noted in the review, there are studies in the literature employing Pt₂CoCu and Pt₂CoNi as oxygen reduction reaction (ORR) electrocatalysts (J. Am. Chem. Soc. 2020, 142, 45; Small, 2023, 19, 2300112). However, these investigations mostly emphasize that the enhanced activity of PtCoM stems from their near-optimum strain. To some extent, these findings also validate the accuracy of our theoretical prediction.

On the other hand, our work provides a novel design perspective by enhancing the ordering of PtCo through element doping. With the aid of DFT and machine learning, we identified Cu and Ni for their capability to increase the thermodynamic driving force for intermetallic formation while preserving the superior intrinsic activity from candidate elements. Experimentally, PtCoCu/Ni (>50% ordering degree) can be achieved at lower annealing temperature and smaller particle size. The prepared ternary alloy PtCoCu/Ni exhibits higher SA compared to PtCo# (small size, low order) and a

larger ECSA than PtCo* (large size, high order). Additionally, enhanced chemical ordering can also effectively contribute to the durability of the catalysts.

We have made the following edit to the text to address the reviewer's concern, and properly cited the corresponding literatures.

[Revision to manuscript] (Page7):

..., and thus were finally selected for the experimental validation. There has been prior research reporting PtCoNi and PtCoCu ternary system as excellent ORR catalyst due to their near-optimum strain levels for higher ORR activity^{33,34}, aligning well with our computational screening results. More importantly, our study provides a novel design perspective that the introduction of Cu/Ni leverages the thermodynamic driving force for the disordered-to-order transition, resulting in a more favorable tradeoff between specific activity (SA) and electrochemically active surface area (ECSA).

3. The advantage or need and configurational materials space of the catalysts were not described. No quantitative metrics of how machine learning accelerated the materials discovery is explained.

Response: We appreciate the reviewer for bringing this to our attention. A detailed response to this question has been provided in the response to the prior question “Does ML is needed to identify M= Ni or Copper?”

4. No GitHub details were provided.

Response: We thank the reviewer for pointing this out. All the necessary data and code have been uploaded to GitHub. Please refer to <https://github.com/ZhangLabTHU/PtCoM> for access to the source data for DFT modeling and machine learning used in this study.

[Revision to manuscript] (Page18):

Data Availability

All source data for DFT modeling and machine learning that were used in this study are available from the GitHub repository: <https://github.com/ZhangLabTHU/PtCoM>

5. There are statements such as PtCoCu or PtCoNi show high ORR activity. No explanations or valid hypothesis were mentioned. Detailed DFT studies are needed to show the binding sites and demonstrate reaction pathways.

Response: Many thanks for the reviewer's suggests on this issue. The microkinetic model proposed by Norskov (J. Phys. Chem. B 2004, 108, 17886-17892) was used to evaluate the ORR activity. A five atomic layers slab model was utilized, and Pt alloy was modeled by three Pt atomic layer supported on two atomic layers composited by Pt₂CoM alloy. We follow the dissociative mechanism which includes oxygen dissociative adsorption (1), formation of OH and desorption of OH coupled with proton and electron transfer (2) and (3), as outlined following:

The adsorption energy of OH and O can be express by:

$$\Delta E_O = E_{O*} - E_* - (E_{H_2O} - E_{H_2}) \quad (4)$$

$$\Delta E_{OH} = E_{OH*} - E_* - (E_{H_2O} - 1/2E_{H_2}) \quad (5)$$

Where E_{O*} , E_{OH*} and E_* are the DFT calculated energies of the O, OH adsorbed surface and clean surface. E_{H_2O} and E_{H_2} are DFT calculated energies of water and hydrogen molecules. The adsorption Gibbs free energy was defined by:

$$\Delta G = \Delta E + \Delta E_{water} + \Delta ZPE - T\Delta S \quad (6)$$

Where ΔE_{water} is the solvation correction, ΔZPE is the zero point correction and T ΔS is the entropic correction, in this work we used correction values from the reference (J. Phys. Chem. B 2004, 108, 17886-17892). The highest potential where all reaction steps are exothermic U_L is defined as:

$$U_L = -\max(\Delta G_1, \Delta G_2, \Delta G_3)/|e^-| \quad (7)$$

Where $\Delta G_1, \Delta G_2, \Delta G_3$ are reaction free energy of reaction (1)~(3).

Fig. R11. (a) Two-dimensional activity volcano plot for ORR activity as a function of OH and O adsorption energy relative to Pt. (b) Free energy diagram for oxygen reduction reaction for Pt, PtCo, Pt₂CoCu, Pt₂CoNi, Pt₂CoGa, Pt₂CoTi at $U = 0$ V, U_L depicts the highest potential where all reaction steps are exothermic. (c) The projected electronic density of states and d band center. (d) The linear relationship between oxygen adsorption energy with surface strain and d band center.

As depicted in Figure R11a, Pt₂CoCu or Pt₂CoNi are the two candidates situated closest to the optimal region in the 2D volcano plot. Figure R11b and Table R3 further

illustrates that the Pt₂CoCu or Pt₂CoNi exhibit the highest limiting potential U_L. Consequently, we claim that Pt₂CoCu or Pt₂CoNi possess highest activity among four kinds of Pt₂CoM.

It is widely acknowledged that when the number of Pt atomic layers exceeds three, the strain effect becomes the dominant factor in regulating the alloy's activity. We calculate the d band center for Pt₂CoM, we also use the surface strain proposed by Yang (Science 374, 459–464, 2021) to quantify the anisotropic strain of Pt₂CoM. The oxygen adsorption energy exhibits a significant linear relationship with surface strain and the d band center, as shown in Figure R1d. Therefore, the enhanced activity of Pt₂CoCu and Pt₂CoNi can be attributed to the strain effect, which regulates the electronic structures of Pt atoms on the surface and optimizes the adsorption energy towards ORR intermediates. Table R4 summarized the DFT ORR optimized intermediates adsorptions configurations. All these additional calculations have been added to the revised Supporting Information.

Table R3: The adsorption energies of O and OH and highest potential values for Pt, order PtCo, Pt₂CoCu, Pt₂CoNi, Pt₂CoGa, Pt₂CoTi and disorder PtCo.

System	ΔE_{OH} (eV)	ΔE_{O} (eV)	U _L (V)
Pt	0.91	1.39	0.66
PtCo (order)	1.06	1.82	0.81
Pt ₂ CoCu	0.94	1.67	0.69
Pt ₂ CoNi	1.01	1.78	0.76
Pt ₂ CoGa	0.94	1.54	0.69
Pt ₂ CoTi	0.91	1.45	0.66
PtCo (disorder)	1.00	1.67	0.79
PtCo (disorder)	0.94	1.66	0.76
PtCo (disorder)	0.93	1.61	0.68
PtCo (disorder)	1.01	1.64	0.69
PtCo (disorder)	1.04	1.66	0.75

Table R4: DFT-optimized ORR intermediates adsorption configurations for order PtCo, Pt₂CoCu, Pt₂CoNi, Pt₂CoGa, Pt₂CoTi and

System	*	* OH	* O
PtCo (order)	 	 	 Pt ₂ CoCu	 	 	 Pt ₂ CoNi	 	 	 Pt ₂ CoGa	 	 	 
Pt₂CoTi			PtCo-1 (disorder)			PtCo-2 (disorder)			PtCo-3 (disorder)			PtCo-4 (disorder)			PtCo-5 (disorder)			

Based on the reviewer's comment, we have added this results in revised Manuscript and Supplementary Information.

[Revision to manuscript]:

1. Page 6:

We calculated the adsorption energies of OH* and O* as the activity descriptor to evaluate the ORR activity of the Pt₂CoM (M= Ga, Ni, Cu and Ti) using Pt-shell slab models with ordered Pt₂CoM-core (*Fig. 2d and Table S3-4*)³². *The computed results show that strain effects play a dominated role in regulating the ORR activity of Pt₂CoM (Fig. S10).*

2. Fig. 2d:

3. (Page13):

The E_{H_2O} and E_{H_2} are energies of H₂O and H₂ are -6.76 eV and -14.22 eV respectively, which are calculated by DFT using a H₂O and H₂ molecule placed on a 10Å×10Å×10Å cubic cell model. *The adsorption Gibbs free energy was defined by:*

$$\Delta G = \Delta E + \Delta E_{water} + \Delta ZPE - T\Delta S \quad (3)$$

Where ΔE_{water} is the solvation correction, ΔZPE is the zero point correction and $T\Delta S$ is the entropic correction, in this work we used correction values from the reference³².

[Revision to Supplementary Information]:

... , **Supplementary Figure 10.** (a) Two-dimensional activity volcano plot for ORR activity as a function of OH and O adsorption energy relative to Pt. (b) Free energy diagram for oxygen reduction reaction for at $U = 0$ V, U_L depicts the highest potential where all reaction steps are exothermic. (c) The projected electronic density of states and d band center. (d) The linear relationship between oxygen adsorption energy with surface strain and d band center.

1. (Page16-19):

... , **Supplementary Table 3.** DFT calculated O and OH adsorption energies and limiting potential for 4 order Pt_2CoM combinations, order PtCo and disorder PtCo.

Supplementary Table 4. Top view and side view of the DFT-optimized ORR intermediates adsorption configurations for order PtCo, Pt_2CoCu , Pt_2CoNi , Pt_2CoGa , Pt_2CoTi and disorder PtCo.

Overall, this reviewer does not recommend this paper for Nature Communications

Response: We hope that all the above point-to-point response have clearly conveyed the novelty of our work, and demonstrate its suitability for publication in Nature Communications. We are also welcome to the review's constructive suggestions to further improve this work.

REVIEWERS' COMMENTS

Reviewer #1 (Remarks to the Author):

The authors addressed my questions well and the revised Ms is in good shape for publication. I have no further comments.